# Ultra-Fine Entity Typing with Prior Knowledge about Labels: A Simple Clustering Based Strategy

**Na Li**
USST, Shanghai
China
li_na@usst.edu.cn

**Zied Bouraoui**
CRIL CNRS & Univ. Artois
France
bouraoui@cril.fr

**Steven Schockaert**
Cardiff University
UK
schockaerts1@cardiff.ac.uk

## Abstract

Ultra-fine entity typing (UFET) is the task of inferring the semantic types, from a large set of fine-grained candidates, that apply to a given entity mention. This task is especially challenging because we only have a small number of training examples for many of the types, even with distant supervision strategies. State-of-the-art models, therefore, have to rely on prior knowledge about the type labels in some way. In this paper, we show that the performance of existing methods can be improved using a simple technique: we use pre-trained label embeddings to cluster the labels into semantic domains and then treat these domains as additional types. We show that this strategy consistently leads to improved results, as long as high-quality label embeddings are used. We furthermore use the label clusters as part of a simple post-processing technique, which results in further performance gains. Both strategies treat the UFET model as a black box and can thus straightforwardly be used to improve a wide range of existing models.[1]

## 1 Introduction

Entity typing is the task of inferring the semantic type(s) of an entity that is mentioned in some sentence. While only a few coarse types were traditionally considered, such as *person* and *organisation*, the focus has shifted to increasingly finer-grained types. Consider the following example:[2] "*The company said its production upgrades would also have a "short-term impact" on the delivery of vaccines to the UK*". From this sentence, we cannot only infer that "the company" refers to an organisation but also that it refers to a pharmaceutical company. To allow for more informative predictions, Ling and Weld (2012) proposed the task of

fine-grained entity typing. However, the 122 entity types that were considered in their work are still too coarse for many applications. For instance, the most specific type from their taxonomy that applies to the example above is *company*. A further refinement of entity types was proposed by Choi et al. (2018), who considered a total of $10,331$ semantic types and introduced the term *ultra-fine entity typing* (UFET). For instance, the ultra-fine type *pharmaceutical* is available for the above example.

UFET is essentially a multi-label text classification problem. The main challenge stems from the fact that we only have a few labelled training examples for most of the semantic types, and sometimes even none at all. The solution proposed by Choi et al. (2018) is based on two distant supervision strategies. Their first strategy is to use mentions that link to a Wikipedia page and use the definition on that page to infer semantic types. Their second strategy relies on nominal expressions which include their semantic type as a head word (e.g. *president Joe Biden*). More recent work has improved on these strategies using denoising techniques (Onoe and Durrett, 2019; Pan et al., 2022).

Even with distant supervision, the number of training examples remains prohibitively small for many types. State-of-the-art methods, therefore, rely on prior knowledge about the labels themselves. For instance, Pan et al. (2022) use the representation of the labels in the decoder of BERT to initialise the weights of the label classifiers, while Huang et al. (2022) use a fine-tuned BERT (Devlin et al., 2019) model to map type labels onto prototypes. However, the token embeddings from BERT are known to be sub-optimal as pre-trained representations of word meaning (Bommasani et al., 2020), whereas fine-tuning a BERT encoder may lead to overfitting. In this paper, we therefore pursue a different strategy to inject prior knowledge about the labels into UFET models. We use pre-trained embeddings to cluster the labels into groups,

---

[1] Our code and evaluation scripts are available at https://github.com/lina-luck/ufet_with_domains.

[2] https://www.bbc.co.uk/news/world-europe-55666399

which we call domains. Rather than changing the UFET model itself, we simply treat these domains as additional semantic types. In other words, given a training example with labels $l_1, ..., l_n$, we now also add the synthetic labels $c_1, ..., c_n$, where $c_i$ is the cluster to which $l_i$ belongs. This strategy has the advantage that arbitrary label embeddings can be used and that the UFET model itself does not need to be modified. Despite its simplicity, we find this strategy to perform remarkably well, provided that high-quality label embeddings are used. The best results are obtained with recent BERT-based strategies for learning word vectors (Li et al., 2023). In contrast, the representations from classical word embedding models (Mikolov et al., 2013; Pennington et al., 2014) are too noisy to have a positive impact, with ConceptNet Numberbatch (Speer et al., 2017) being a notable exception.

Apart from using pre-trained label embeddings, another line of work has focused on modelling label correlations (Jiang et al., 2022). Essentially, the idea is to train a probabilistic model on top of the predictions of a base UFET classifier. While meaningful improvements are possible in this way, the extent to which label correlations can be learned is inherently limited due to the sparsity of the training data. As an alternative, we focus on two simple strategies to improve the predictions of the base UFET model. Both strategies directly rely on prior knowledge about the labels and are thus well-suited for labels that are rare in the training set. The first strategy is aimed at inferring missing labels: if the domain label $c_i$ is predicted but none of the labels that belong to that domain, we add the most likely label from $c_i$ to the predicted set. The second strategy is aimed at removing conflicting labels: if an entity is predicted to have multiple labels from the same cluster, then we use a pre-trained classifier to check whether these labels are mutually inconsistent, and if so, we remove the labels with the lowest confidence degree.

The contributions of this paper can be summarised as follows. We use pre-trained label embeddings to group the given set of type labels into semantic domains. These domains are then used to improve a base UFET model in different ways:

- By adding the domains as synthetic labels to the training examples (Section 4), the UFET model is exposed to the knowledge that these labels have something in common.

- After the UFET model is trained, we can use

the domains to post-process its predictions (Section 5). In this way, we can infer missing labels or remove conflicting labels.

## 2 Related Work

**Ultra-Fine Entity Typing** Methods for ultra-fine entity typing typically rely on training sets that are automatically constructed from Wikipedia links (Ling and Weld, 2012; Choi et al., 2018). As a different strategy, Dai et al. (2021) extract weakly labelled examples from a masked language model, using Hearst-like patterns (Hearst, 1992) to rephrase the original sentence. For instance, the sentence *"In late 2015, Leonardo DiCaprio starred in The Revenant"* is transformed into *"In late 2015, [MASK] such as Leonardo DiCaprio starred in The Revenant"*. The predictions of the language model for the [MASK] token are then used as weak labels for the target entity. Yet another possibility, considered by Li et al. (2022), relies on transfer learning. Specifically, they use a pre-trained Natural Language Inference (NLI) model, where the original sentence is used as the premise, and the hypothesis is a sentence expressing that the target entity has a given type. While their approach achieves strong results, they have to run the NLI model for every possible type, which is prohibitively expensive for ultra-fine entity typing.

**Label Dependencies** In fine-grained entity typing, there are usually various dependencies between the labels. For instance, in the case of FIGER (Ling and Weld, 2012) and OntoNotes (Gillick et al., 2014), the labels are organised into a tree, which models can exploit (Ren et al., 2016; Shimaoka et al., 2017; Xu and Barbosa, 2018; Murty et al., 2018; Chen et al., 2020). In the case of UFET, however, the label set is essentially flat (Choi et al., 2018), which means that label dependencies have to be learned from the training data and/or external sources. Xiong et al. (2019) use a Graph Neural Network (GNN) (Kipf and Welling, 2017) to take label correlations into account. The edges of the graph, which indicate potential label interactions, are selected based on (i) label co-occurrence in the training data and (ii) the similarity of the GloVe embeddings of the label names. The model thus relies on the assumption that labels with similar embeddings are likely to apply to the same entities, which is too strong: the embeddings of mutually exclusive labels, such as *teacher* and *student*, can also be similar. Liu et al. (2021b) rely on two strate-

gies for modelling label correlation. One strategy is based on directly observed label correlations. For the other strategy, they replace the entity by the [MASK] token and obtain predictions from BERT for this token. Then they model correlations between these predictions and the actual labels. Li et al. (2021a) also use a GNN to model label interactions. The nodes of their graphs correspond to both entity types and keywords from the given sentences. Jiang et al. (2022) model label correlations using a probabilistic graphical model.

Different from the aforementioned works, we exploit prior knowledge about label dependencies in a way which is decoupled from the entity typing model, by adding synthetic labels to training examples and by post-processing the predictions. This makes our method more transparent (i.e. we know which dependencies are used and why) and modular (i.e. our method can directly be applied to existing entity typing models). While the lack of integration with the entity typing model may seem overly simplistic, note that for most entity types we only have a few training examples. It is difficult to learn meaningful label dependencies from such data without also learning spurious correlations.

**Label Embeddings** Recent UFET models typically rely on some kind of prior knowledge about the labels. As already mentioned, Xiong et al. (2019) rely on GloVe embeddings to construct a label graph. Pan et al. (2022) initialise the scoring function for each label based on that label's representation in the decoder of the language model (see Section 3). Rather than using pre-trained embeddings, Huang et al. (2022) encode the labels using a language model, which is fine-tuned together with the entity mention encoder. A similar approach was used by Ma et al. (2022) in the context of few-shot entity typing. In these approaches, the label embeddings are directly used as label prototypes for the classification model, which could make it harder to distinguish between similar but mutually exclusive labels. In a broader context, label embeddings have also been used for few-shot text (Hou et al., 2020; Halder et al., 2020; Luo et al., 2021) and image (Xing et al., 2019; Yan et al., 2021, 2022) classification. Clusters of label embeddings have been used in the context of extreme multi-label text classification (Prabhu et al., 2018; Chang et al., 2020; Jiang et al., 2021). However, in this setting, clusters are used to make the problem tractable, rather than being a strategy for injecting prior knowledge.

The representation of each label is typically derived from the sparse features of the documents that label, and the focus is on learning balanced clusters.

## 3 Problem Setting

In ultra-fine entity typing, the input consists of a sentence in which a particular entity mention is highlighted. The aim is to assign all the semantic types that apply to that entity, given a large set $\mathcal{L}$ of around 10,000 possible types (Choi et al., 2018). This task can be treated as a standard multi-label classification problem. In particular, let $\phi$ be an encoder which maps an input sentence $s$ with entity mention $m$ to a corresponding embedding $\phi(s, m) \in \mathbb{R}^n$. Together with the encoder $\phi$, we also learn a scoring function $f_l : \mathbb{R}^n \rightarrow [0, 1]$ for each label $l \in \mathcal{L}$. The probability that label $l$ applies to the entity mentioned in the input is then estimated as $f_l(\phi(s, m))$. As a representative recent approach, we consider the DenoiseFET model from Pan et al. (2022), which uses a BERT encoder with the following input:

$$s\,[P_1]\,m\,[P_2][P_3][\text{MASK}]$$

Here $s$ is the input sentence, $m$ is the span corresponding to the entity mention, and $[P_1], [P_2], [P_3]$ are trainable tokens. These trainable tokens intuitively allow the model to learn a soft prompt, and were found to improve performance by Pan et al. (2022). The encoding $\phi(s, m)$ is then taken to be the embedding of [MASK] in the final layer of the BERT model. The scoring functions $f_l$ take the form $f_l(\mathbf{x}) = \sigma(\mathbf{x} \cdot \mathbf{y}_l + b_l)$, where $\sigma$ is the sigmoid function, $\mathbf{y}_l \in \mathbb{R}^n$ corresponds to a prototype of label $l$ and $b_l \in \mathbb{R}$ is a bias term. The parameters $\mathbf{y}_l$ and $b_l$ are initialised according to the corresponding weights in the decoder of the masked language model (where the average across all tokens is used if $l$ consists of several tokens).

## 4 Adding Domain Labels

We now describe our main strategy for improving UFET models based on pre-trained label embeddings. Broadly speaking, the label embeddings provide us with prior knowledge about which labels are similar. Crucially, however, the fact that two labels have similar embeddings does not necessarily mean that the occurrence of these labels is positively correlated. Labels can be similar because they denote categories with similar meanings (e.g. *student* and *learner*), but similar labels can

also denote mutually exclusive categories from the same domain (e.g. *student* and *teacher*, or *ambulance* and *fire truck*). For this reason, we do not require that the scoring functions (or prototypes) for similar labels should always be similar. Instead, we propose a strategy based on clusters of labels.

We use an off-the-shelf clustering method to cluster labels based on their pre-trained embedding. Each cluster intuitively represents a *domain*. We associate a synthetic label with each of the domains and add these synthetic labels to the training examples. For instance, consider the following cluster:

$$\mathcal{L}_i = \{\textit{fire truck}, \textit{fire engine}, \textit{air ambulance},$$
$$\textit{ambulance}, \textit{police car}\} \qquad (1)$$

Whenever a training example has any of these labels, we add "cluster $i$" as an additional label. This tells the model that examples about different types of emergency vehicles all have something in common.[3] If a linear scoring function is used, as in DenoiseFET, examples involving emergency vehicles are thus encouraged to be linearly separated from other examples. This helps the model to uncover features that are common to emergency vehicles, while not requiring that prototypes of different types of emergency vehicles be similar.

# 5 Post-processing Predictions

Domains can also be used for improving the predictions of an UFET model *post hoc*. We focus on two simple strategies that exclusively rely on prior knowledge about the label dependencies. In particular, we do not attempt to learn about label dependencies from the training data. We assume that dependencies between frequent labels are already modelled by the base UFET model, whereas for rare labels, relying on the training data may lead to overfitting. This also has the advantage that we can continue to treat the base UFET model as a black box. We only require that the base UFET model can provide us with a confidence score $\text{conf}(l; s, m)$ for any label $l \in \mathcal{L}$ and input $(s, m)$.

## 5.1 Inferring Missing Labels

The use of domain labels allows us to improve the predictions of the model through a simple post-processing heuristic. Let $\mathcal{L}_1, ..., \mathcal{L}_k$ be the different domains, and let us write $c_i$ for the synthetic domain label that was introduced to describe cluster

$\mathcal{L}_i$. If $c_i$ is predicted for a given input $(s, m)$, then it seems reasonable to assume that we should also predict at least one label from $\mathcal{L}_i$. Indeed, the meaning of $c_i$ was precisely that one of the labels from $\mathcal{L}_i$ applies. If $c_i$ is predicted but none of the labels from $\mathcal{L}_i$, we therefore add the label $l$ from $\mathcal{L}_i$ with the highest confidence score $\text{conf}(l; s, m)$ to the set of predicted labels. For instance, consider again the cluster $\mathcal{L}_i$ from (1). If $c_i$ is predicted but none of the labels in $\mathcal{L}_i$, then this intuitively means that the model believes the entity is an emergency vehicle, but it has insufficient confidence in any of the individual types in $\mathcal{L}_i$. However, its confidence in *ambulance* may still be higher than its confidence in the other options, in which case we add *ambulance* to the set of predicted labels. Note that this is clearly a recall-oriented strategy.

## 5.2 Identifying Conceptual Neighbourhood

As we already discussed, the labels in a given cluster often correspond to mutually exclusive categories from the same domain. We will refer to such labels as *conceptual neighbours*.[4] For instance, the same entity cannot be both an ambulance and a police car, even though these entity types are similar. Now suppose we have prior knowledge about which labels are conceptual neighbours. We can then implement a straightforward post-processing method: if the model predicts a label set which contains two conceptual neighbours, we should only keep one of them. In such a case, we simply discard the label in which the model was least confident.

We now discuss how knowledge about conceptual neighbourhood can be learned. Bouraoui et al. (2020) proposed a method for predicting whether two BabelNet (Navigli and Ponzetto, 2010) concepts are conceptual neighbours. Their method can only be used for concepts whose instances are named entities (for which we have a pre-trained embedding). Hence we cannot use their strategy directly. However, using their method, we obtained a list of 5521 positive examples (i.e. concept pairs from BabelNet which are believed to be conceptual neighbours) and 2318 negative examples. We then trained a classifier to detect conceptual neighbourhood. We started from a Natural Language Inference (NLI) model that was pre-trained on the WANLI dataset (Liu et al., 2022), initialised from

---

[3] Note that the label "emergency vehicle" is not included in the vocabulary of the UFET dataset from (Choi et al., 2018).

[4] Formally, conceptual neighbours are concepts that are represented by adjacent regions in a conceptual space (Gärdenfors, 2000). In this paper, however, we treat the notion of conceptual neighbourhood in a more informal fashion.

RoBERTa-large (Liu et al., 2019), and we fine-tuned this model using the training examples from BabelNet. Specifically, for a concept pair $(l_1, l_2)$ we use *"The category is $l_1$"* as the premise and *"The category is $l_2$"* as the hypothesis. We train the model to predict *contradiction* for positive examples and *neutral* for negative examples (as $l_1$ and $l_2$ are always co-hyponyms in the training examples).

To test the effectiveness of the conceptual neighbourhood classifier, we carried out an experiment where 20% of the BabelNet training data was held out. After training the Conceptual Neighbourhood classifier on the remaining 80% of the training data, we achieved an accuracy of 97.2% on the held-out fragment, suggesting that the considered strategy is indeed highly effective.

Once the NLI model is fine-tuned, we use it to identify conceptual neighbourhood among the labels in $\mathcal{L}$. To this end, we test every pair of labels from the same domain. The label pairs that are classified as positive examples are then used for implementing the considered post-processing strategy. Note that this is a precision-oriented strategy.

## 6 Experimental Results

We evaluate our approach on the ultra-fine entity typing (UFET) benchmark from Choi et al. (2018). Although we expect our strategies to be most effective for ultra-fine entity types, we also carry out an evaluation on two standard fine-grained entity typing benchmarks: OntoNotes[5] (Gillick et al., 2014) and FIGER[6] (Ling and Weld, 2012).

### 6.1 Experimental Set-up

**Pre-trained Label Embeddings**  We consider several pre-trained label embeddings. First, we include a number of standard word embedding models: Skip-gram[7] (Mikolov et al., 2013), GloVe (Pennington et al., 2014)[8], SynGCN (Vashishth et al., 2019), Word2Sense (Panigrahi et al., 2019), and ConceptNet Numberbatch (Speer et al., 2017). Beyond traditional word embeddings, we also include two models that distil static word embeddings from BERT-based language models: MirrorBERT (Liu et al., 2021a) and the model from Gajbhiye et al.

---

[5] http://nlp.cs.washington.edu/entity_type/data/ultrafine_acl18.tar.gz
[6] https://www.cs.jhu.edu/~s.zhang/data/figet/Wiki.zip
[7] We used the model that was trained on Google News (https://code.google.com/archive/p/word2vec/).
[8] We used the model that was trained on Common Crawl (https://nlp.stanford.edu/projects/glove/).

| Method | LM | P | R | F1 |
|---|---|---|---|---|
| Box4Types* | Bl | 52.8 | 38.8 | 44.8 |
| LRN* | Bb | 54.5 | 38.9 | 45.4 |
| MLMET* | Bb | 53.6 | 45.3 | 49.1 |
| DenoiseFET* | Bb | **55.6** | 44.7 | 49.5 |
| UNIST* | Rb | 49.2 | 49.4 | 49.3 |
| UNIST* | Rl | 50.2 | 49.6 | 49.9 |
| NPCRF* | Bb | 52.1 | 47.5 | 49.7 |
| NPCRF* | Bl | 55.3 | 46.7 | 50.6 |
| LITE* | Rl$_{\text{MNLI}}$ | 52.4 | 48.9 | 50.6 |
| DenoiseFET | Bb | 52.6 | 46.2 | 49.2 |
| DenoiseFET | Bl | 52.6 | 47.5 | 49.8 |
| DenoiseFET | Rb | 52.3 | 46.0 | 49.0 |
| DenoiseFET | Rl | 52.9 | 47.4 | 50.0 |
| DenoiseFET + PKL | Bb | 52.7 | 49.2 | 50.9 |
| DenoiseFET + PKL | Bl | 53.8 | **50.2** | **51.9** |
| DenoiseFET + PKL | Rb | 53.1 | 48.5 | 50.7 |
| DenoiseFET + PKL | Rl | 53.7 | 50.1 | 51.8 |

Table 1: Results for Ultra-Fine Entity Typing (UFET), in terms of macro-averaged precision, recall and F1. Results with * are taken from the original papers. The LM encoders are BERT-base (Bb), BERT-large (Bl), RoBERTa-base (Rb), RoBERTa-large (Rl) and RoBERTA-large fine-tuned on MNLI (Rl$_{\text{MNLI}}$).

(2022), which we will refer to as ComBiEnc. The former is a contrastively fine-tuned BERT-base encoder, which maps any given word onto a static vector. The latter is a BERT-base encoder that was fine-tuned to predict commonsense properties. Finally, we include two approaches which learn word embeddings by averaging the contextualised representations of different mentions: the AvgMASK method from (Li et al., 2021b) and the ConCN method from (Li et al., 2023). The former obtains the representation of a word $w$ by finding up to 500 mentions of that word in Wikipedia. These mentions are masked, and the contextualised representations of the MASK token are obtained using RoBERTa-large. The resulting vectors are finally averaged to obtain the embedding of $w$. ConCN follows a similar strategy, but instead of a pre-trained RoBERTa model, they use a RoBERTa-large model that was contrastively fine-tuned using distant supervision from ConceptNet. We use this latter model as our default choice for the experiments in Section 6.2. A comparison with the other embeddings is presented in Section 6.3.

**Training Details**  For the conceptual neighbourhood classifier, we tune the confidence level above which we consider two labels to be conceptual neighbours using the development set. We use Affinity Propagation (AP) as the clustering algo-

| Method | LM | OntoNotes | | FIGER | |
|---|---|---|---|---|---|
| | | macro | micro | macro | micro |
| Box4Types* | Bl | 77.3 | 70.9 | 79.4 | 75.0 |
| LRN* | Bb | 84.5 | 79.3 | - | - |
| MLMET* | Bb | 85.4 | 80.4 | - | - |
| DenoiseFET* | Bb | 87.1 | 81.5 | - | - |
| NPCRF* | Bb | 85.2 | 80.0 | - | - |
| NPCRF* | Rl | 86.0 | 81.9 | - | - |
| DSAM* | LSTM | 83.1 | 78.2 | 83.3 | 81.5 |
| SEPREM* | Rl | - | - | 86.1 | 82.1 |
| RIB* | ELMo | 84.5 | 79.2 | **87.7** | **84.4** |
| LITE* | Rl$_{MNLI}$ | 86.4 | 80.9 | 86.7 | 83.3 |
| DenoiseFET | Bb | 87.2 | 81.4 | 86.2 | 82.8 |
| DenoiseFET | Bl | 87.5 | 81.6 | 86.5 | 82.9 |
| DenoiseFET | Rb | 87.4 | 81.5 | 86.4 | 82.9 |
| DenoiseFET | Rl | 87.6 | 81.8 | 86.7 | 83.0 |
| DenoiseFET + PKL | Bb | **87.7** | **81.9** | 86.8 | 82.9 |
| DenoiseFET + PKL | Bl | 87.9 | 82.1 | 87.0 | 83.1 |
| DenoiseFET + PKL | Rb | 87.8 | 82.2 | 86.9 | 83.0 |
| DenoiseFET + PKL | Rl | 87.9 | 82.3 | 87.1 | 83.1 |

Table 2: Results for fine-grained entity typing, in terms of macro-F1 and micro-F1. Results with * are taken from the original papers. The LM encoders are BERT-base (Bb), BERT-large (Bl), RoBERTa-base (Rb), RoBERTa-large (Rl) and RoBERTA-large fine-tuned on MNLI (Rl$_{MNLI}$).

rithm. Affinity Propagation does not require us to specify the number of clusters, but instead requires a so-called preference value to be specified. Rather than tuning this preference value, we obtain clusters for each of the following values: $\{0.5, 0.6, 0.7, 0.8, 0.9\}$. We use all these clusters together. This means that our method essentially uses domains at multiple levels of granularity. We use DenoiseFET (Pan et al., 2022) as the base entity typing model. For UFET and OntoNotes, we also rely on their denoised training set[9]. For FIGER, we use the standard training set.

## 6.2 Results

We refer to our proposed strategy as PKL (Prior Knowledge about Labels). The results for UFET are presented in Table 1. Following the original paper, we report macro-averaged precision, recall and F1. We compare our model with Box4Types (Onoe et al., 2021), LRN (Liu et al., 2021b), MLMET (Dai et al., 2021), DenoiseFET (Pan et al., 2022), UNIST (Huang et al., 2022), NPCRF (Jiang et al., 2022) and LITE (Li et al., 2022). The baseline results were obtained from the original papers. For each method, we also mention which pre-trained

[9]Available from https://github.com/CCIIPLab/DenoiseFET.

| Method | P | R | F1 |
|---|---|---|---|
| DenoiseFET + DL + missing + CN | 52.7 | **49.2** | **50.9** |
| DenoiseFET + DL + missing | 52.9 | 48.5 | 50.6 |
| DenoiseFET + DL | 52.5 | 48.4 | 50.4 |
| DenoiseFET + LLE | **53.0** | 46.1 | 49.4 |
| DenoiseFET | 52.6 | 46.2 | 49.2 |

Table 3: Ablation analysis on UFET for different variants of our model, using BERT-base for the entity encoder.

| Embedding | P | R | F1 |
|---|---|---|---|
| Skip-gram | 53.9 | 44.8 | 49.0 |
| GloVe | 53.5 | 45.9 | 49.4 |
| Numberbatch | 53.5 | 46.9 | 50.0 |
| Word2Sense | 53.7 | 45.5 | 49.3 |
| SynGCN | **54.6** | 44.3 | 48.9 |
| MirrorBERT | 51.7 | 46.9 | 49.2 |
| ComBiEnc | 53.2 | 47.9 | 50.4 |
| AvgMASK | 53.7 | 46.9 | 50.1 |
| ConCN | 52.7 | **49.2** | **50.9** |

Table 4: Performance of different pre-trained label embeddings on UFET, using DenoiseFET with BERT-base as the base UFET model.

model was used for initialising the entity encoder. As can be seen, our approach outperforms all baselines. This is even the case when using BERT-base or RoBERTa-base for the entity encoder, although the best results are obtained with BERT-large.

The results for fine-grained entity typing are presented in Table 2. Following the tradition for these benchmarks, we report both macro-averaged and micro-averaged F1. In addition to the baselines from Table 1 (for which results for OntoNotes or FIGER are available), we also include results for the following fine-grained entity typing models: DSAM (Hu et al., 2021), RIB (Li et al., 2021a) and SEPREM (Xu et al., 2021). For both OntoNotes and FIGER, our model consistently outperforms the base DenoiseFET model. While the improvements are smaller than for UFET, this is nonetheless remarkable, given that our strategies are targeted at ultra-fine entity types. For OntoNotes, our model outperforms all baselines. In the case of FIGER, its performance is comparable to LITE (i.e. slightly better macro-F1 but worse micro-F1) and outperformed by RIB. The latter model uses a GNN based strategy to learn correlations between labels and keywords from the training data.

## 6.3 Analysis

We now present some additional analysis, to better understand which components of the model are

| Input | Predictions |
|---|---|
| Jazz coach Jerry Sloan, angry over the officiating, had two technicals called against **him** and was ejected from the game in the closing minutes. | person, coach, *trainer*, ~~athlete~~, ~~player~~ |
| Queen Mary, the Queen Mother, came to see **it**, and after the performance asked Cotes where he found "those wonderful actors." | play, performance, show, *event*, ~~opera~~ |
| Eurostat said output fell in all euro nations where **it** had data, with Finland booking the sharpest drop at 23.2 per cent | organization, institution, corporation *agency*, *administration*, ~~company~~, ~~business~~, ~~firm~~ |

Table 5: Examples of differences between the predictions of the base model and those of the full model, using BERT-base. Labels in green are only predicted by the full model; labels in red are only predicted by the base model.

| Input | Predictions | Gold standard |
|---|---|---|
| Yes – although regular coffee drinking isn't harmful for most people, that might not hold true for **pregnant women**. | person, adult, female, *patient*, woman | person, adult, female, woman |
| **They**, like the other children, are listed as missing. | person, child, *son* | person, child |
| During **the American Revolutionary War** he served as secretary to Admiral Molyneux Shuldham, in Boston in 1776 and again at Plymouth (1777-78). | event, conflict, battle, *fight*, war | time, event, conflict, battle, era, struggle, war |
| Matlock almost immediately formed his own band, **Rich Kids**, with Midge Ure, Steve New, and Rusty Egan. | group, *organization*, musician, band | group, musician, band |
| As **the War of the Fourth Coalition** broke out in September 1806, Emperor Napoleon I took his Grande Armee into the heart of Germany in a memorable campaign against Prussia. | event, conflict, battle, ~~dispute~~, fight, struggle, war | event, conflict, battle, dispute, fight, struggle, war, competitiveness, group-action |
| During the Second World War, Ailsa built vessels for the Navy, including **several Bangor class minesweepers**. | object, ship, boat, craft, ~~vessel~~ | object, ship, boat, thing, vessel |
| "If you get your pitch, and take a good swing, anything can happen," **he** later remarked. | person, athlete, ~~coach~~, player | person, coach, player, trainer |

Table 6: Error analysis, showing cases where the full model incorrectly added or removed a label (compared to the base model), using BERT-base. Labels in green are only predicted by the full model; labels in red are only predicted by the base model.

most responsible for its outperformance.

**Ablation Study** In Table 3, we report the results that are obtained when only using some of the components of our strategy. We use DL to refer to the use of domain labels (Section 4), *missing* to refer to post-processing based on domain labels (Section 5.1) and *CN* for the post-processing based on conceptual neighbourhood (Section 5.2). We also include a variant that uses Local Linear Embeddings (LLE) as an alternative strategy for taking into account the pre-trained label embeddings. LLE acts as a regularisation term in the loss function, which essentially imposes the condition that prototypes with similar labels should themselves also be similar. As we have argued in Section 4, this is not always appropriate. A detailed description of how

we use LLE is provided in the appendix. While adding LLE to the base UFET model has a small positive effect, we can see that the performance is much worse than that of our proposed clustering based strategy (DL). The two post-processing techniques (missing and CN) both have a positive effect on the overall performance, although they have a smaller impact than the DL strategy.

**Label Embedding Comparison** In Table 4, we compare the performance of different pre-trained label embeddings. In all cases, we use our full model with BERT-base for the entity encoder. The results show that the choice of word embedding model has a substantial impact on the result, with ConCN clearly outperforming the traditional word embedding models. Overall, we might expect that

| Label | Conceptual Neighbours |
|---|---|
| hip | knee, thigh, ankle, shoulder-joint |
| high-school | intermediate-school, middle-school, junior-school, junior-high |
| liver-cancer | stomach-cancer, brain-cancer, lung-cancer |
| maple | cherry-wood, oak, cedar |
| pink | purple, blue, red, yellow |
| quarterly | monthly, yearly, weekly |
| rabbit | silver-fox, duck, sheep |
| mouth | throat, nose, ear |
| baht | ringgit, yen, yuan |
| native-bear | sun-bear, bear-cat, cave-bear |
| pickup-truck | monster-truck, station-wagon, estate-car |
| red-wine | white-wine, table-wine, ice-wine |
| second-baseman | third-baseman, first-baseman, left-fielder, right-fielder |
| summer-camp | holiday-camp, work-camp, labour-camp |
| bar-chart | pie-chart, box-plot, heat-map |
| dinner-jacket | evening-dress, morning-dress, dress |

Table 7: Examples of conceptual neighbours for some labels from the UFET benchmark.

word embeddings distilled from language models capture word meaning in a richer way. From this perspective, the strong performance of Numberbatch also stands out. Interestingly, ConCN, ComBiEnc and Numberbatch all rely on ConceptNet, which suggests that having label embeddings that capture commonsense properties might be important for achieving the best results.

**Qualitative Analysis** Table 5 shows some examples where the post-processing strategies improve the predictions of the base model (DenoiseFET). On the one hand, our method results in a number of additional labels being predicted. On the other hand, the conceptual neighbourhood strategy removes some incompatible labels. In the first example, *athlete* and *player* are removed because they are conceptual neighbours of *coach*; in the second example, *opera* is removed because it is a conceptual neighbour of *play*; in the third example, *company*, *business* and *firm* are removed because they are conceptual neighbours of *corporation*.

Table 6 shows a number of examples where the full model has introduced a mistake, compared to the base model. In the top four examples, the full model predicted an additional label that was not included in the ground truth. As can be seen, the added labels are somewhat semantically close, especially in the last two cases, where it could be argued that the labels are actually valid. In the bottom three examples, the conceptual neighbourhood strategy incorrectly removed a label. In particular, in the first of these, *dispute* was removed because

| | Domains |
|---|---|
| **GloVe** | {grape, red wine, rice wine, white wine, wine, champagne, ice wine, port wine} |
| | {bag, luggage, sack, plastic bag, purse, shoulder bag} |
| | {reception, desk, help desk, lobby, reception desk} |
| | {ban, banning, moratorium, prohibition} |
| | {knee, leg, ankle, elbow, hamstring, knee cap, thigh} |
| **Numberbatch** | {outlook, perspective, view, viewing, vista} |
| | {image, photo, photograph, picture, portrait} |
| | {digit, finger, hand, thumb, toe, index finger, ring finger} |
| | {approval, licence, license, permission, permit, authorization, consent, planning permission} |
| | {artificial intelligence, computer vision, expert system, machine learning, speech recognition, voice recognition} |
| **MirrorBERT** | {permit, residence, residence permit, work permit} |
| | {century, decade, era, period, generations, millennium} |
| | {analysis, assessment, evaluation, measurement, appraisal} |
| | {career criminal, hate crime, serial killer, war crime, war criminal, capital crime} |
| | {doctor, medical, patient, physician, psychiatrist, surgeon} |
| **ConCN** | {social, social media, social network, social networking, twitter, facebook, google, social life, video chat, yahoo, youtube} |
| | {change, transformation, transition, sea change, sex change, step change, transform, alteration} |
| | {acre, inch, square, square foot, square inch, square meter, square mile} |
| | {client, customer, customer base, clientele} |
| | {beef, flesh, meat, pork, ground beef, lamb, mystery meat, red meat} |

Table 8: Examples of the domains that were found with different word embeddings.

it was (incorrectly) predicted to be a conceptual neighbour of *conflict*. In the following example, *vessel* was incorrectly assumed to be a conceptual neighbour of *ship*. In the final example, *coach* was removed as a conceptual neighbour of *player*. Unlike the previous two cases, it is less clear whether this is actually a mistake. Indeed, it is highly unlikely that the person who is referred to is both a coach and a player. However, the sentence leaves it ambiguous which of these labels applies. In general, for DenoiseFET with BERT-large on UFET, the *missing* strategy from Section 5.1 was applied to 744 out of 1998 test instances. A total of 879

labels were added, and 513 of these were correct. The conceptual neighbourhood strategy affected 504 instances. A total of 983 labels were removed, and 709 of these deletions were correct.

Table 7 lists some examples of conceptual neighbours that were predicted by our classifier (for the UFET label set). As can be seen, in most cases, the conceptual neighbours that were found indeed refer to disjoint concepts. However, we can also see a small number of false positives (e.g. *summer camp* and *holiday camp*).

Finally, Table 8 shows examples of domains that were found. As can be seen, most of the domains are meaningful. However, in the case of Mirror-BERT, we also find clusters containing terms with rather different meanings, such as *permit* and *residence* in the first example. Often such terms are thematically related but taxonomically distinct. For instance, in the last cluster, we find two sub-types of *person*, namely *career criminal* and *serial killer*, together with sub-types of *crime*. Treating these labels as being part of the same domain may confuse the model, as they apply to very different types of entities. For GloVe, we can similarly see words which are thematically related but taxonomically different. This is illustrated in the first example, where *grape* is included in a cluster about wines.

## 7 Conclusions

We have analysed the effectiveness of two types of external knowledge for ultra-fine entity typing: pre-trained label embeddings and a conceptual neighbourhood classifier. The strategies that we considered are straightforward to implement and can be added to most existing entity typing models. One of the most effective strategies simply requires adding synthetic labels, corresponding to clusters of semantic types, to the training examples. We also discussed two post-processing strategies which can further improve the results. Despite their simplicity, the proposed strategies were found to consistently improve the performance of the base UFET model. The nature of the pre-trained label embeddings played an important role in the performance of the overall method. An interesting avenue for future work would be to design strategies for identifying label domains which are specifically designed for the UFET setting.

## Limitations

Using conceptual neighbourhood for post-processing the predictions of an entity typing model is straightforward and effective, but there are other important types of label dependencies which we are currently not considering. Entailment relations, for instance, could be learned in a similar way to how we learn conceptual neighbourhood. In some cases, it could also be useful to combine the strategies we have proposed in this paper with models that learn label correlations from the training data itself, especially for fine-grained (rather than ultra-fine grained) entity types. However, as we have argued in the paper, this would come with the risk of overfitting the training data. The label domains that we currently use are essentially taxonomic categories, i.e. sets of labels that have similar meanings (e.g. *student* and *teacher*). We may expect that incorporating thematic groupings of labels could further improve the results (e.g. where *student* would be grouped with other university related terms such as *lecture hall*).

**Acknowledgments** This work was supported by EPSRC grant EP/V025961/1, ANR-22-CE23-0002 ERIANA and by HPC resources from GENCI-IDRIS (Grant 2023-[AD011013338R1]). Na Li is supported by Shanghai Big Data Management System Engineering Research Center Open Funding.

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

## A   Locally Linear Embedding

Locally Linear Embedding (LLE) (Roweis and Saul, 2000) is a well-known dimensionality reduction technique, which aims to preserve linear structure in the neighbourhood of each data point. Rather than using LLE for dimensionality reduction, we use it for regularising the space of the label prototypes. This is somewhat reminiscent of how LLE has been used in NLP for transforming word embeddings (Hasan and Curry, 2017).

Let us write $N_l$ for the $k$ nearest labels to $l \in \mathcal{L}$, in terms of their cosine similarity in the pre-trained embedding space. We can approximate the embedding $\mathbf{l}$ as a linear combination of the embeddings of the labels in $N_l$, by solving the following optimisation problem:

$$\text{Minimise} \left( \mathbf{l} - \sum_{p \in N_l} w_{lp}\mathbf{p} \right)^2 \quad \text{s.t.} \sum_{p \in N_l} w_{lp} = 1$$

Then we add the following regularisation term to the loss function of the entity typing model:

$$\mathcal{L}_{\text{LLE}} = \sum_{l \in \mathcal{L}} \left( \mathbf{y}_l - \sum_{p \in N_l} w_{lp}\mathbf{y}_p \right)^2$$

In other words, we encourage the model to preserve the linear dependence that holds between $l$ and

its neighbours in the pre-trained embedding space. The overall loss then becomes:

$$\mathcal{L} = \mathcal{L}_{\text{entity}} + \lambda \mathcal{L}_{\text{LLE}}$$

where $\mathcal{L}_{\text{entity}}$ is the loss function from the entity typing model and $\lambda > 0$ is a hyper-parameter to control the strength of the regularisation term.