# OpenReview forum: "Ultra-Fine Entity Typing with Prior Knowledge about Labels: A Simple Clustering Based Strategy"
_EMNLP/2023/Conference — EMNLP 2023 Findings_

### Official Review · Reviewer_h56F · 2023-08-02

**Soundness:** 4

**Excitement:**

3: Ambivalent: It has merits (e.g., it reports state-of-the-art results, the idea is nice), but there are key weaknesses (e.g., it describes incremental work), and it can significantly benefit from another round of revision. However, I won't object to accepting it if my co-reviewers champion it.

**Paper Topic And Main Contributions:**

What this paper about?
- This paper addresses Ultra-Fine Entity Typing (UFET).

What problem or question did this paper address?
- The problems is that the number of training examples remains prohibitively small for many of the types (Line 62-64).

What are the main contributions that this paper makes towards a solution or answer?
- The authors use pre-trained label embeddings to group the given set of type labels into semantic domains. (Line 121-123).


**Reasons To Accept:**

- The proposed method contributes to performance improvement (Tables 1 and 2).
- The ablation study shows to what extent each element of the proposed method contributes to performance improvement.


**Reasons To Reject:**

- While the proposed method improves the performance, the improvement is modest (about 1.0 F1 score).

**Reproducibility:**

4: Could mostly reproduce the results, but there may be some variation because of sample variance or minor variations in their interpretation of the protocol or method.

**Reviewer Confidence:**

3: Pretty sure, but there's a chance I missed something. Although I have a good feel for this area in general, I did not carefully check the paper's details, e.g., the math, experimental design, or novelty.

---

> ### Author Rebuttal · Authors · 2023-08-25
>
> We thank the reviewer for their careful review.
>
> Please note our response to the first reviewer (QZ94) regarding the point about the improvement being modest.

---

### Official Review · Reviewer_AV7U · 2023-08-04

**Soundness:** 3

**Excitement:**

3: Ambivalent: It has merits (e.g., it reports state-of-the-art results, the idea is nice), but there are key weaknesses (e.g., it describes incremental work), and it can significantly benefit from another round of revision. However, I won't object to accepting it if my co-reviewers champion it.

**Paper Topic And Main Contributions:**

This paper studies the task of ultra-fine entity typing (UFET). They propose a methodology that takes advantage of the similarities of labels through conceptual neighborhoods. The propose to cluster the labels into domains that constitute of dissimilar but related labels, eg., fire truck, fire engine, air ambulance, ambulance and police car. They include the cluster labels (aka domains) to the existing label space and present a post-hoc filter of labels from a black-box UFET model.

The paper outperforms the state-of-the-art methods on the UFET dataset from Choi et al., 2018. It also includes a qualitative analysis of the utility of proposed cluster labels.

**Questions For The Authors:**

1. What is the intuition behind special tokens P1, P2 and P3? (236–237)
2. What's a prototype of a label? y_l in lines 241, 242
3. It is not clear how the clustering method ensures each cluster consists of mutually exclusive categories (322--324). How accurate is the clustering method?
4. While predicting conceptual neighborhoods using the NLI model, how are conflicts resolved in pairwise decisions? For example, transitivity might not hold between the three labels, fire truck, fire engine, and police car.
5. How accurate is the NLI model used for conceptual neighbor prediction? both on the BabelNet test set and the domains from the UFET dataset.
6. Add a short explanation for the local linear neighborhood (474--476).
7. How to compute embeddings for cluster labels?


**Reasons To Accept:**

1. The paper is well-motivated. They make progress toward capturing semantic relatedness among the labels from the large label space of UFET task. This shows a resemblance to hierarchical typing and can provide useful insights for future work on this task.
2. The ablation studies highlight the effectivness of each proposed module, clustering labels, post-processing to detect missing and conflicting labels from conceptual neighborhoods.

**Reasons To Reject:**

1. In its current form, it's hard to follow the methods described in the paper. The paper omits a number of details that are necessary to understand the paper (see questions below).
2. The paper suggests the effectiveness of proposed methods to the long-tail of entity types. However, it doesn't include any empirical results to showcase this abilities. As I understand it, the baselines already work very well for high-resource types, so it would be important to see how the use of cluster labels benefits the long-tail.
3. The paper can also benefit from a comparison against prior work on modeling label dependencies. The related work section briefly highlights this, however there is not empirical comparison on the label dependencies captured in prior work and current work.

**Reproducibility:**

3: Could reproduce the results with some difficulty. The settings of parameters are underspecified or subjectively determined; the training/evaluation data are not widely available.

**Reviewer Confidence:**

4: Quite sure. I tried to check the important points carefully. It's unlikely, though conceivable, that I missed something that should affect my ratings.

---

> ### Author Rebuttal · Authors · 2023-08-25
>
> We thank the reviewer for their careful review.
>
> As suggested by the reviewer, we will add an analysis that focuses on ultra-fine entity types in particular. For instance, when using BERT-base as the encoder, if we only consider the ultra-fine types in the UFET benchmark, the performance increases from 34.6 for the base model to 38.5 for our model.
>
> The reviewer suggests comparing against prior work on modeling label dependencies. In fact, such a comparison is already made in the paper. For instance, NPCRF uses neural conditional random fields to model label dependencies, while LRN and RIB use graph neural networks for this purpose.
>
> Q1: The special tokens are supposed to represent a learnable soft prompt. Including such tokens was found to improve performance in previous work, and is now a standard approach in work on UFET.
>
> Q2: We use the term “prototype” to refer to the vector y_l, since the label l is predicted for an embedding x based on the similarity between x and y_l.
>
> Q3: We do not guarantee that the labels in a cluster are mutually exclusive. We simply make the empirical observation that this is often the case. Note that we use the conceptual neighbourhood classifier to verify when this is actually the case (i.e. if the clustering algorithm were to guarantee mutual exclusivity then we would not actually need the conceptual neighbourhood classifier).
>
> Q4: We start by removing the label with the lowest confidence (which is involved in at least one conflict). This process is repeated until all conflicts are resolved.
>
> Q5: We have now carried out an experiment where 20% of the BabelNet training data was held out. After training the Conceptual Neighbourhood classifier on the remaining 80% of the training data, we achieved an accuracy of 97.2% on the held-out fragment. This confirms the insights from our qualitative analysis, which also suggested that the conceptual neighbourhood classifier is very accurate. We will add this analysis to the paper.
>
> Q6: Note that LLE is explained in Appendix A, while the intuition behind LLE is explained on lines 476-480.
>
> Q7: Note that we only need embeddings of cluster labels for the LLE baseline. In that case, we simply represent a cluster by averaging the embeddings of the labels that belong to that cluster.

---

### Official Review · Reviewer_QZ94 · 2023-08-05

**Soundness:** 4

**Excitement:**

3: Ambivalent: It has merits (e.g., it reports state-of-the-art results, the idea is nice), but there are key weaknesses (e.g., it describes incremental work), and it can significantly benefit from another round of revision. However, I won't object to accepting it if my co-reviewers champion it.

**Paper Topic And Main Contributions:**

This paper proposes a fine-grained entity typing system that adds pseudo-labels representing a cluster of actual labels to the training and inference process. This way, the recall can be improved, especially on the low-occurring types. The authors propose two post-hoc processing constraints associated with the pseudo-labels: 1) if a pseudo-label is predicted, at least one actual type from of the cluster represented by the pseudo-label should be predicted; 2) if an actual type is predicted, then none of its mutually-exclusive types should be predicted. Such mutually exclusive types are called conceptual neighbors and are learned with a contrastive NLI model.

Experiments show that the proposed suite of methods that aim to inject prior knowledge regarding each type into the process collectively improve the baseline model's performances on the UFET dataset and marginally on two other fine-grained entity typing benchmarks. Ablation studies also support the claim that each technique is helpful in the overall performance.

**Reasons To Accept:**

The proposed suite of method effectively injects prior knowledge regarding fine-grained types into the entity-typing process. Because of the pseudo-labels, low-occurring types can be better covered during both training and post-hoc inference.

**Reasons To Reject:**

1) The improvement is marginal and incremental, and the paper does not touch or discuss the core problem that leads to the low-performances of the UFET dataset.


**Reproducibility:**

3: Could reproduce the results with some difficulty. The settings of parameters are underspecified or subjectively determined; the training/evaluation data are not widely available.

**Reviewer Confidence:**

4: Quite sure. I tried to check the important points carefully. It's unlikely, though conceivable, that I missed something that should affect my ratings.

---

> ### Author Rebuttal · Authors · 2023-08-25
>
> We thank the reviewer for their careful review.
>
> For ultra-fine entity typing, the proposed approach improves the corresponding base models by around 2 percentage points (e.g. from 49.8 to 51.9 in the case of BERT-large). While this may appear like a small improvement, it represents a more substantial gain than what has been reported in other recent work in this area. We believe this is an important finding, especially given how easy it is to apply our proposed methods.
>
> Moreover, when we focus on the finest-grained entity types (i.e. the ones designated as ultra-fine in the UFET dataset), the improvement is larger. If we only consider such types, the F1 score improves from 34.6 for the base model to 38.5 for our model (when using BERT-base as the encoder). We will add an analysis focusing only on ultra-fine types to the paper to make this clearer.

---

### Meta-Review · Area_Chair_9TQb · 2023-09-19

**Recommendation:** 4

**Metareview:**

The paper proposes a fine-grained entity typing system using pseudo-labels to enhance recall for less frequent types. Post-hoc constraints are introduced for these labels to ensure prediction of at least one actual type from the cluster and avoid prediction of mutually-exclusive types. These exclusive types, termed conceptual neighbors, are learned using a contrastive NLI model. Enhancements in the paper's presentation are warranted. The experimental results should explicitly demonstrate the system's superior performance, particularly emphasizing its effectiveness with respect to long-tail types.

---

### Decision · Program_Chairs · 2023-10-07

**Decision:**

Accept-Findings

**Comment:**

The paper proposes a fine-grained entity typing system using pseudo-labels to enhance recall for less frequent types. Post-hoc constraints are introduced for these labels to ensure prediction of at least one actual type from the cluster and avoid prediction of mutually-exclusive types. These exclusive types, termed conceptual neighbors, are learned using a contrastive NLI model. Enhancements in the paper's presentation are warranted. The experimental results should explicitly demonstrate the system's superior performance, particularly emphasizing its effectiveness with respect to long-tail types.